# A Fish-like Binocular Vision System for Underwater Perception of Robotic Fish

**DOI:** 10.3390/biomimetics9030171

**Published:** 2024-03-12

**Authors:** Ru Tong, Zhengxing Wu, Jinge Wang, Yupei Huang, Di Chen, Junzhi Yu

**Affiliations:** 1Laboratory of Cognitive and Decision Intelligence for Complex System, Institute of Automation, Chinese Academy of Sciences, Beijing 100190, China; tongru2019@ia.ac.cn (R.T.); zhengxing.wu@ia.ac.cn (Z.W.); huangyupei2021@ia.ac.cn (Y.H.); 2School of Artificial Intelligence, University of Chinese Academy of Sciences, Beijing 100049, China; 3Human-Oriented Methodology and Equipment Laboratory, Department of Advanced Manufacturing and Robotics, College of Engineering, Peking University, Beijing 100871, China; wangjinge@stu.pku.edu.cn; 4State Key Laboratory for Turbulence and Complex Systems, Department of Advanced Manufacturing and Robotics, College of Engineering, Peking University, Beijing 100871, China; di.chen@pku.edu.cn

**Keywords:** fish-like vision system, underwater perception, underwater robot, field of view stitching, robotic fish

## Abstract

Biological fish exhibit a remarkably broad-spectrum visual perception capability. Inspired by the eye arrangement of biological fish, we design a fish-like binocular vision system, thereby endowing underwater bionic robots with an exceptionally broad visual perception capacity. Firstly, based on the design principles of binocular visual field overlap and tangency to streamlined shapes, a fish-like vision system is developed for underwater robots, enabling wide-field underwater perception without a waterproof cover. Secondly, addressing the significant distortion and parallax of the vision system, a visual field stitching algorithm is proposed to merge the binocular fields of view and obtain a complete perception image. Thirdly, an orientation alignment method is proposed that draws scales for yaw and pitch angles in the stitched images to provide a reference for the orientation of objects of interest within the field of view. Finally, underwater experiments evaluate the perception capabilities of the fish-like vision system, confirming the effectiveness of the visual field stitching algorithm and the orientation alignment method. The results show that the constructed vision system, when used underwater, achieves a horizontal field of view of 306.56°. The conducted work advances the visual perception capabilities of underwater robots and presents a novel approach to and insight for fish-inspired visual systems.

## 1. Introduction

In recent years, autonomous underwater vehicles have been continuously developed and have a wide range of applications in underwater searches and environmental monitoring. The demand for underwater vehicles with visual perception has drawn attention to the study of underwater computer vision. Although some underwater robots are capable of carrying visual devices to capture images and videos, research in underwater vision remains a relatively under-explored field [1].

In underwater environments, visual perception has advantages over common sonar imaging [2], as reflected in being cost-effective, feature-rich, and containing semantic information. Visual perception is widely used in underwater vehicles [3]. For example, Huang et al. focused on improving the operational precision of the end-effector system of underwater robots through visual servo control [4]. This research demonstrates that uncalibrated visual perception, guided by reinforcement learning, can direct the robot to perform repeatable actions. Visual perception, when used for control feedback, requires high frequency and low latency, and hence, in underwater robots, it is more commonly employed for target identification and tracking, navigation, and positioning [5].

Identifying and tracking underwater equipment such as cables and pipelines is a common task for underwater robots with vision perception [6,7]. Onboard vision systems detect targets in visual images by using feature points or lines in order to achieve tracking and inspection. A robot navigation system that combines visual data with acoustic data can provide the relative spatial position of the cable, achieving autonomous cable tracking and inspection [8]. In the task of underwater fiber-optic cable inspection, existing solutions face limitations in the field of view, often requiring the tracking strategies to compensate for the shortcomings in visibility [7].

In terms of positioning, visual odometry is a natural technological approach and is often used for self-localization of underwater robots [9]. Research on underwater visual odometry aids with solving precise positioning and navigation in unstructured underwater environments. Furthermore, facing the harshly changing underwater environment, Wang et al. combined depth information with 2D visual images to obtain continuous and robust self-localization information [10]. The integration of underwater visual positioning systems with inertial measurement devices enhances accuracy, offering a cost-effective alternative to expensive acoustic positioning solutions. This is a one of the key factors in the growing interest in underwater visual systems. In recent years, underwater visual simultaneous localization and mapping (SLAM) has also been a focus of research, particularly in terms of precise positioning and mapping [11].

The visual system is crucial for enhancing the autonomy of underwater robots and is now used in advanced tasks such as the recovery of AUVs in shallow water [12], docking of underwater vehicles [13], and hitchhiking of bionic robotic fish [14]. Visual perception is the fundamental base for expanding the application fields of underwater robots. However, current research on the visual systems of underwater robots still faces the following difficulties. First, image calibration is complex. In underwater environments, it is necessary to consider the refraction of the water medium and to establish a refraction model for the waterproof cover. This makes camera calibration complex, and the refraction from the waterproof cover results in loss of field of view [15]. Second, the range of visual perception is limited. The narrow underwater perceptual field of view can cause missing of targets, often necessitating additional strategies to compensate for the constraints of the narrow visual range [7]; however, multi-camera systems can expand the field of view to improve the success rate of target tracking [13]. A wider or even panoramic field of view is beneficial for the efficiency and accuracy of underwater searches or target tracking, but this usually requires a redundant number of cameras. Third, the movement of underwater robots can cause jitter, making the stability of the visual system a consideration. Especially in bionic robot systems, vision stabilization methods are necessary [16].

Imitating human eye perception and with an understanding of vision, human-like stereovision has been designed and used for robust perception in vehicles [17]; in underwater environments, inspired by the fish eyes of biological fish, fish-like vision systems are easier to deploy on robots with fish-like streamlined shapes and can achieve a wider field of view at a lower cost.

A fish-like vision system is a type of binocular system characterized by a minimal overlap region and severe distortion. The depth calculation of conventional binocular cameras is not effective in the minimal overlap regions of the fish-like vision system. However, research on binocular visual field stitching algorithms can be conducted to obtain continuous, ultra-wide-range perception images, which can enhance the practical application value of the fish-like vision systems. Binocular visual field stitching combines images with overlapping regions to form wide-field and high-resolution images. Its main steps include feature matching, image registration, and seam removal [18]. Due to the constraints of adverse visual environments, there are fewer feature points and a higher matching error rate, which lead to difficulties in underwater image stitching [19]. Although improvements in natural feature extraction and matching can improve the accuracy of underwater stitching [19,20,21], they are more commonly used on image sequences for which the adjacent images themselves have a higher degree of overlap. Leveraging the characteristic of unchanged relative positions of cameras, some binocular-vision-based methods have significantly enhanced processing efficiency and robustness compared to traditional stitching algorithms that rely solely on image appearance information [22,23]. However, past research has often utilized binocular cameras with parallel optical axes and has relied on accurate stereo calibration. The latter poses challenges for fish-like vision systems with minimal overlap regions and severe distortion. In the post-processing stage of the registration, seam cutting can produce visually appealing stitched images from partially aligned images [24]. Various optimization methods based on colors [25], edges [26], depths [27], and other features have been studied to adapt to different scenes.

Considering the current state of underwater vision research, this paper presents a fish-like binocular vision system for underwater robots. Compared to common underwater vision systems, it is expected to obtain a wide field of view perception through a reasonable combination of two fisheye cameras. The contributions of this paper primarily include the following three aspects. Firstly, a fish-like binocular vision system is designed and implemented and features a structure adapted to the streamlined shape of the robot and biomimetic field of view characteristics. Through the field of view design method proposed, the fish-like vision system can be successfully deployed on robotic fish. Secondly, in consideration of the characteristics of significant distortion and disparity in the visual system, a field of view stitching method is proposed to obtain complete perceptual images, and the field of view of the stitched images is tested and showed a maximum field of view reaching 306.56°. Thirdly, with the assistance of a calibration board, an orientation alignment method is employed to draw orientation indicators in stitched images, providing reference for the localization and tracking of targets within the field of view by underwater robots.

The rest of this paper is organized as follows. Section 2 presents the fish-like vision system, corresponding visual field design method, and deployment process. In Section 3, a visual field stitching method is proposed for merging images from two fisheye cameras and obtaining complete perceptual images. In Section 4, an orientation alignment method is proposed for drawing yaw and pitch scales within the field of view. Section 5 describes the experimental tests. Finally, the conclusions are summarized and future works are presented in Section 6.

## 2. Fish-like Binocular Vision System

The research objective of the fish-like vision system is twofold: on the one hand, it aims to obtain a larger perceptual field of view with as minimal visual hardware as possible, primarily by mimicking the positional distribution of biological fish eyes; on the other hand, it seeks to provide a vision system solution that has minimal field loss and a large field of view while also being adapted to the streamlined shape of the bionic robotic fish. Therefore, we develop a fish-like vision system as shown in Figure 1, which mainly includes two parts: a design method for a wide field of view adapting to streamlined shapes and a deployment method without waterproof compartments.

### 2.1. Field of View Design

The fish-like vision system needs to adapt to irregular streamlined shapes on the one hand and achieve the widest possible field of perception on the other hand. Based on the design requirements, a field of view design method is proposed that considers three key elements, as shown in Figure 2.

Optical axis perpendicular to the tangent plane: On the premise of initially replicating the shape of a robotic fish, the fisheye camera is placed in a position mimicking that of a biological fish eye. The protective lens of the fisheye camera is directly and tightly integrated with the streamlined shape of the robot and does not rely on additional waterproof covers. The waterproof-cover-free solution not only prevents attenuation of the camera’s field of view but also replicates the central position of the biological fish’s eye as closely as possible. To minimize the loss of the streamlined shape of the robotic fish, it is necessary for the optical axis to be perpendicular to the tangent plane of the streamlined shape. To satisfy the tangency condition, the position of the camera can be represented by two angles (ψ,τ). Specifically, the left and right cameras rotate clockwise and counterclockwise by ψ/2 respectively from facing directly left and right and then rotate by τ around the camera’s horizontal axes $l_{1}$ and $l_{2}$ respectively. Since the camera is placed tangentially to the streamlined shape, (ψ,τ) can also be used to describe the streamlined shape features at the installation location.

Binocular visual field overlap: In biological fish, the intersection of the fields of view of both eyes is usually small, and in some cases, there is virtually no intersection. For the fish-like vision system, the two fisheye cameras correspond to the two eyes of a biological fish, and the binocular visual field overlap [28] is beneficial for forming a continuous and complete field of view. Therefore, in the field of view design, there should be a certain degree of overlap angle φ between the two eyes’ fields of view; this is typically around 10° and need not be excessively large.

As large a field of view as possible: In underwater scenarios, a large field of view is beneficial for robots to capture more information, and reducing the blind spots of the binocular vision system is expected to enhance the efficiency of autonomous tasks such as underwater searching and inspection. The discussion on the field of view design is focused on the field of view design plane, as shown by the plane $\Pi_{\zeta }$ in Figure 2. On the design plane $\Pi_{\zeta }$, the overlapping projected area of the two fisheye cameras is maximized, and the angle of overlap on this plane is defined as φ. When the field of view angle of the fisheye camera is θ, the range of the field of view can be represented as Θ=2θ−φ. Therefore, the condition for maximizing the field of view is expressed as follows:(1)$$\begin{array}{c} \max2\theta -\varphi \end{array}$$

To further clarify the relationship between the field of view, the streamlined shape, and the camera mounting angle, the field of view design geometric model in Figure 2 is established, which thereby allows us to design a field of view that meets the desired expectations. First, the mathematical symbols in Figure 2 are clarified.
$$\begin{array}{cc} l_{1} & \;\mathrm{The}\;\mathrm{intersection}\;\mathrm{line}\;\mathrm{of}\;\mathrm{the}\;\mathrm{left}\;\mathrm{camera}\;\mathrm{plane}\;\mathrm{with}\;\mathrm{the}\;\mathrm{horizontal}\;\mathrm{plane}.\\ l_{2} & \;\mathrm{The}\;\mathrm{intersection}\;\mathrm{line}\;\mathrm{of}\;\mathrm{the}\;\mathrm{right}\;\mathrm{camera}\;\mathrm{plane}\;\mathrm{with}\;\mathrm{the}\;\mathrm{horizontal}\;\mathrm{plane}.\\ \psi & \;\mathrm{The}\;\mathrm{angle}\;\mathrm{between}\;l_{1}\;\mathrm{and}\;l_{2}.\\ \tau & \;\mathrm{The}\;\mathrm{angle}\;\mathrm{of}\;\mathrm{rotation}\;\mathrm{along}\;\mathrm{the}\;l_{1}\;\mathrm{and}\;l_{2}\;\mathrm{axes}\;\mathrm{when}\;\mathrm{installing}\;\mathrm{the}\;\mathrm{camera}.\;\mathrm{The}\;\mathrm{angles}\;\\ \; & \;\mathrm{of}\;\mathrm{rotation}\;\mathrm{are}\;\mathrm{the}\;\mathrm{same}\;\mathrm{for}\;\mathrm{both}\;\mathrm{cameras}\;\mathrm{but}\;\mathrm{are}\;\mathrm{in}\;\mathrm{opposite}\;\mathrm{directions}.\\ \Pi_{\zeta } & \;\mathrm{The}\;\mathrm{field}\;\mathrm{of}\;\mathrm{view}\;\mathrm{design}\;\mathrm{plane}\;\mathrm{with}\;\mathrm{the}\;\mathrm{maximum}\;\mathrm{extent}\;\mathrm{of}\;\mathrm{binocular}\;\mathrm{overlap}.\\ \Pi_{l} & \;\mathrm{The}\;\mathrm{outer}\;\tan\mathrm{gent}\;\mathrm{plane}\;\mathrm{of}\;\mathrm{the}\;\mathrm{streamlined}\;\mathrm{shape}\;\mathrm{at}\;\mathrm{the}\;\mathrm{camera}\;\mathrm{mounting}\;\mathrm{location}.\\ \varphi & \;\mathrm{The}\;\mathrm{degree}\;\mathrm{of}\;\mathrm{overlap}\;\mathrm{angle}\;\mathrm{between}\;\mathrm{two}\;\mathrm{eyes}.\\ \xi & \;\mathrm{The}\;\mathrm{angle}\;\mathrm{of}\;\mathrm{the}\;\mathrm{direction}\;\mathrm{of}\;\mathrm{maximum}\;\mathrm{overlapping}\;\mathrm{in}\;\mathrm{the}\;\mathrm{image},\\ \; & \;i.e.,\;\mathrm{the}\;\mathrm{angle}\;\mathrm{between}\;\mathrm{the}\;\mathrm{intersection}\;\mathrm{line}\;\mathrm{of}\;\mathrm{the}\;\mathrm{camera}\;\mathrm{plane}\;\mathrm{and}\;\mathrm{the}\;\mathrm{design}\;\mathrm{plane},\;\\ \; & \;\mathrm{and}\;\mathrm{the}\;\mathrm{mounting}\;\mathrm{axes}\;l_{1},\;l_{2}.\\ \zeta & \;\mathrm{The}\;\mathrm{angle}\;\mathrm{between}\;\mathrm{the}\;\mathrm{field}\;\mathrm{of}\;\mathrm{view}\;\mathrm{design}\;\mathrm{plane}\;\Pi_{\zeta }\;\mathrm{and}\;\mathrm{the}\;\mathrm{horizontal}\;\mathrm{plane}. \end{array}$$

The field of view design plane is formed by the directions of maximum overlapping of the left and right fisheye camera views. This plane has two characteristics: firstly, the binocular visual images on the field of view design surface are continuous; thus, the images from both eyes can be stitched along this direction; secondly, the angle of overlap is the largest on this plane, allowing the field of view Θ on this plane to be used as the measurement standard for the visual field range of the fish-like vision system.

Furthermore, by analyzing the geometric features, the relationship between the field of view design surface angle ζ and the angle ξ of the maximum overlap direction in the image with the angles (ψ,τ) of the streamlined shape can be obtained. Firstly, (ψ,τ) determine the angle of the tangent plane $\Pi_{l}$. Generally, the streamlined shape is symmetrical, so when considering the tangent plane of the left eye alone, its rotational characteristics can be represented by (ψ/2,τ). The tangent plane of the left eye is derived by rotating the vertical plane around the z-axis by ψ/2 and then around the x-axis by τ. To facilitate the analysis of geometric features, a simplified diagram of the geometric relationships between planes and axes is depicted in Figure 2, with all geometric relationships contained within the tetrahedron $O_{h}O_{1}O_{2}O_{c}$. The three colors in Figure 2 refer to elements on the three planes, respectively. Analyzing the simplified geometric diagram, $O_{h}O_{c}$ is the tangent to $l_{o1}$ and $l_{o2}$, *C* is the projection of $O_{c}$ onto the horizontal plane, and $Q_{1}$ and $Q_{2}$ are the foot points on planes $\Pi_{l}$ and $\Pi_{\zeta }$, respectively. The symbol $\tau_{-}$ refers to the angle between the tangent plane $\Pi_{l}$ with the horizontal plane $\Pi_{h}$, where $\tau_{-}=90^{\circ }-\tau$. The symbol ζ refers to the angle between the design plane $\Pi_{\zeta }$ and the horizontal plane $\Pi_{h}$. Therefore, based on the radius *R* of the image plane circle, the lengths of the sides can be represented as
(2)$$\begin{array}{cc} \; & \left|O_{h}O_{c}\right|=R\tan\xi ,\;\left|O_{h}O_{1}\right|=R/\cos\xi ,\;\left|O_{h}Q_{1}\right|=R/\cos\xi -R\cos\xi \\ \; & \left|O_{C}Q_{1}\right|=\left|O_{h}Q_{1}\right|\tan\left(90^{\circ }-\xi \right),\;\left|CQ_{1}\right|=\left|O_{h}Q_{1}\right|\tan\left(\psi /2\right)\\ \; & \left|O_{h}Q_{2}\right|=\left|O_{h}O_{1}\right|\cos\left(\psi /2\right) \end{array}$$

According to the trigonometric relationships, the following equations hold in the right triangles $O_{C}Q_{1}C$ and $O_{C}Q_{2}C$:(3)$$\begin{array}{c} \cos\tau_{-}=\left|CQ_{1}\right|/\left|O_{C}Q_{1}\right|,\;\sin\zeta =\left|O_{h}O_{c}\right|/\left|O_{h}Q_{2}\right| \end{array}$$

Combining Equations (Equation 2) and (Equation 3), the relationship between the angles ψ,τ,ζ, and ξ is as follows:(4)$$\begin{array}{cc} \cot\left(\xi \right) & =\tan\left(\psi /2\right)/\sin\left(\tau \right)\\ \sin\left(\zeta \right) & =\sin\left(\xi \right)/\cos\left(\psi /2\right)\\ \Theta & =2\theta -\varphi \;\left(\mathrm{on}\;\mathrm{the}\;\mathrm{plane}\;\Pi_{\zeta }\right) \end{array}$$

Therefore, the field of view characteristics of the fish-like vision system can be described as (ζ,ξ,Θ), the streamlined shape or installation angles are represented by (ψ,τ), and the camera parameters are denoted by θ. Among these, the field of view characteristics (ζ,ξ,Θ) affect the visual perception range. A larger ζ indicates a visual perception tendency towards observing upper regions, a smaller ξ means the overlapping perception area is more focused in the forward direction, and a larger Θ signifies a larger observational field of view.

According to Formula (Equation 4), on the one hand, the system’s field of view characteristics can be calculated based on known streamlined shapes and camera parameters; on the other hand, the field of view characteristics can be designed based on observational task requirements; thereby, users can select the required camera characteristics and adjust the streamlined shape accordingly.

### 2.2. Deployment on Robotic Fish

The fish-like vision system is designed without a waterproof cover; when deployed on a biomimetic robotic fish, it requires the design of a connector that fits the streamlined shape and a reliable sealing solution. The deployment process is shown in Figure 3.

To adapt to the streamlined shape of the bionic robotic fish, at the selected optical axis position, a ring area fitting the streamlined shape is cut out to serve as the base curved surface for the connector. The inner diameter of the ring area matches the outer diameter of the lens, and the ring area has a certain width. Based on this structure, an incremental expansion forms the connector, as shown in Figure 3. The connector is tightly connected to the camera, and the outer curved surface is sealed at the connection interface. Subsequently, the connector is firmly connected to the shell of the streamlined shape with screws.

In this work, the built vision system is specifically installed on a type of bionic robotic tuna that is intended for underwater search tasks. A 210° ultra-wide-angle fisheye camera is chosen, and the angle of the overlapping area set to around 20°. The underwater robotic fish platform deploying the fish-like vision system is shown in Figure 3, and subsequent image algorithms are also deployed on this platform.

## 3. Binocular Visual Field Stitching for Fish-like Vision System

The fish-like vision system is suitable for underwater applications of bionic robotic fish: not only does it conform to the streamlined shape of the fish, but it also significantly increases the range of the perceptual field of view. However, when applied to higher-level algorithms such as target recognition, separately processing the left and right eye images may result in errors, such as incomplete detection of targets or redundant counting of targets in binocular overlap regions, as shown in Figure 4.

The image stitching method can merge binocular images to obtain a continuous and complete image, effectively avoiding ambiguity in processing left and right images. However, the fish-like vision system has the characteristics of large distortion and large disparity, posing certain challenges to the stitching of left and right fields of view. This section proposes a binocular visual field stitching method for a fish-like vision system with large disparity and distortion, and the process is illustrated in Figure 5.

The stitching algorithm transforms the original images from the left and right eyes ($I_{\mathrm{origin},\mathrm{left}},I_{\mathrm{origin},\mathrm{right}}$) into a field of view stitched image ($I_{\mathrm{stitch}}$). For fisheye cameras with significant distortion, the notion of panoramic stitching can be adopted, where multiple fisheye lenses are arranged to obtain a panoramic projection image. For stitching scenes with large disparity, a method based on seam lines can be utilized [24]; it does not require perfect alignment of two images but achieves visually appealing stitching by key joining at the seam lines. The binocular visual field stitching algorithm mainly includes the following four steps.

(1) Camera calibration: Fisheye cameras exhibit significant distortion, and pinhole camera calibration algorithms may not yield accurate results. For fisheye cameras, OCamCalib [29,30,31] provides a convenient means to obtain the parameters for the two fisheye cameras. This process requires the capture of checkerboard pattern images from two cameras.

(2) Feature point extraction and matching: The underwater environment is complex and constantly changing and has poor lighting conditions and limited availability of natural features. This poses challenges for feature detection and matching in the overlapping regions of binocular images. Attempts have been made to produce calibrated images using classic feature detection methods such as SIFT, SURF, ORB, as well as the deep-learning-based SuperPoint feature detection method [32], but obtaining a sufficient and accurate set of feature point matches has proved to be difficult, which makes image registration and stitching difficult to perform. Therefore, for the proposed fish-like visual system, a marker-assisted feature-enhanced matching method was designed.

First, we introduce positioning markers such as ARUCO [33] and patterned markers like chessboards into the overlapping region to enhance the feature points within the area, as shown in Figure 5. Unlike natural feature points, these artificial marker features are specially designed and are easier to detect. Secondly, by detecting the ARUCO marker, we obtain the placement directions $o_{\mathrm{left}}$ and $o_{\mathrm{right}}$ of the artificial markers and simultaneously detect chessboard feature points in the left and right views. Then, we number the incomplete chessboard points in the left and right views. In the left view, numbering starts from the (0,0) point relative to $o_{\mathrm{left}}$ in terms of pixel distance and direction. In the right view, numbering starts from the (0,n) point relative to $o_{\mathrm{right}}$ in terms of pixel distance and direction, with the maximum number reaching (m,n), where m and n represent the length and width, respectively, of the corner point array within the chessboard grid. Finally, points with the same numbering are matched feature point pairs in the left and right views, as shown in the blue-numbered region in Figure 5.

In natural underwater scenarios, features are sparse. The proposed feature matching method uses low-cost chessboard markers, which are easier to make and obtain compared to specially designed three-dimensional markers [34,35]. The proposed marker-assisted feature-enhanced matching method avoids time-consuming and laborious underwater scene setting and provides the feature point pairs needed for image stitching in a cost-effective manner. Through the marker-assisted feature-enhanced matching method, the difficulties of feature detection and matching in the underwater environment are addressed.

(3) Image stitching: Based on the feature point pairs between the left and right views, we compute the homography matrix H using RANSAC [36], and we subsequently calculate the projection relationship. Let *w* be the image width, and $\left(u_{1},v_{1}\right)$ and $\left(u_{2},v_{2}\right)$, respectively, represent the average pixel coordinates of all matched feature points in the two images. The following coordinate relationships can be obtained:(5)$$\begin{array}{c} {\left(u_{1},v_{1},1\right)}^{T}\approx H{\left(u_{2}+w,v_{2},1\right)}^{T} \end{array}$$
To center the field of view, we left-multiply both sides by $2\left(H^{-1}+I\right)/2$, yielding $H_{1}=\left(H^{-1}+I\right)/2$ and $H_{2}=\left(H+I\right)/2$, which are used to simultaneously transform the left and right images. To achieve a stitching result wherein the feature points overlap as much as possible, further optimization of some camera intrinsic and extrinsic parameters is necessary for re-projection. We assuming the position of each camera remains unchanged, while slight rotations along the X,Y,Z axes are permissible, and the focal lengths $f_{1}$ and $f_{2}$ of the cameras can vary within a certain range. We use the quasi-Newtonian method to minimize the following function:(6)$$\begin{array}{c} L=\sum_{i}∥p_{i}-q_{i}∥ \end{array}$$
where $p_{i}$ and $q_{i}$ are the re-projection vectors of the *i*-th pair of pixel feature points after parameter adjustment. We save the optimized camera parameters and homography matrices $H_{1}$ and $H_{2}$ so that subsequent real-time stitching of binocular images can be performed without relying on calibration boards.

(4) Image optimization: Optimization of the stitched images consists primarily of two steps: color correction and seam cutting. Color optimization is achieved by histogram equalization of the image color to ensure color consistency in the stitched field of view. Based on pixel differences [25], we compute an energy map of the overlapping region to locate the seam line, and then, inspired by [37], we implement SSIM-based seam evaluation [38], misaligned component extraction, local patch alignment, and seam merging to improve the smoothness and prevent having a noticeable seam line in the stitched image.

## 4. Orientation Alignment for Fish-like Vision System

For underwater robots, visual perception plays a vital role in applications such as underwater searches and facility maintenance. When tracking underwater targets based on a visual system, the orientation information of hot-spot targets in the field of view can be fed back from the visual image, providing reference data for the robot’s tracking motion. For a bionic robotic fish, the orientation information mainly includes the heading angle and pitch angle. Therefore, we design an orientation alignment method for the fish-like vision system and draw the yaw and pitch indicators on the stitched image $I_{\mathrm{stitch}}$, as shown in Figure 6.

In an underwater environment, both the robotic fish and the calibration board are placed horizontally, with the orientation of the robotic fish parallel to the chessboard grid. The center of gravity of the fish body, $o_{c}$ (closer to the camera), is horizontally aligned with a corner point *o* on the chessboard grid at a distance of *d*. In the scenario depicted in Figure 6, the left eye is first oriented towards the chessboard grid, and visual images from the left and right eyes are captured. After stitching, the left eye’s alignment image $I_{\mathrm{stitch},\mathrm{left}}$ is obtained. Subsequently, the robotic fish is rotated so that the right eye camera faces the chessboard grid. Visual images of the left and right eyes are again captured, and after stitching, the right eye’s alignment image $I_{\mathrm{stitch},\mathrm{right}}$ is obtained. Corner detection is then performed on the registered images of the left and right eyes and acquires the pixel coordinates $p_{\left(i,j\right)}$ corresponding to the physical position coordinates of the chessboard corners $P_{\left(i,j\right)}$. The actual position of the corner point $P_{\left(i,j\right)}$ is represented by the grid distance from point *o*; for example, for the point P shown in Figure 6, i=2, j=3. The angles between the line connecting $P_{\left(i,j\right)}$ and $o_{c}$ with the horizontal and vertical planes are α and β, respectively, which are related to the yaw and pitch angles of point *P* relative to the robotic fish.
(7)$$\begin{array}{cc} \tan\alpha =\frac{i\cdot d_{c}}{d}, & \;\gamma =-\alpha \\ \tan\beta =\frac{j\cdot d_{c}}{d}, & \;\delta_{\mathrm{left}}=90^{\circ }+\beta \\ \; & \;\delta_{\mathrm{right}}=\beta -90^{\circ } \end{array}$$
where $d_{c}$ represents the side length of the chessboard grid, and γ and δ are the pitch and yaw angles, respectively, in the robotic fish’s coordinate system. Assuming point $o_{c}$ is close to the camera, the targets along the $o_{c}$ direction approximately corresponds to the pixel coordinates $p_{\left(i,j\right)}$ in the stitched image. Ultimately, using the pixel coordinate points and their corresponding yaw and pitch angles isopleths, the yaw and pitch scales are drawn on the stitched images.

## 5. Experiments and Results

The fish-like vision system is deployed on a bionic robotic fish platform and employs two 210° industrial cameras connected via USB cables in order to capture underwater image data. The primary data captured includes three categories: stitched calibration image data, totaling 12 sets; orientation alignment images, totaling 8 sets; and video stream data, totaling 3 sets. The underwater experiments are performed from four aspects: field of view test, visual field stitching test, orientation indicator test, and comprehensive performance test of the fish-like vision system in order to verify the perception ability and image algorithm effects.

### 5.1. Field of View Test

The fish-like vision system is capable of obtaining a wide perceptual field of view with just two cameras, and it is necessary to assess the specific size of the field of view through testing. In the constructed fisheye vision system, τ = 29.15°, Ψ = 25.7°; theoretically, *φ* = 55.7 °, *θ* = 210°. Thus, the visual system’s field-of-view characteristic angles ξ = 64.91° and ζ = 68.26° are calculated based on Equation (Equation 4), and theoretically, Θ = 364.3°. However, due to the attenuation of the camera’s field-of-view angle θ and the binocular overlap angle φ in underwater scenes, the confidence level of the theoretical calculation value of the field-of-view range Θ is relatively low.

To accurately measure the perception range of the fish-like vision system, an underwater scene is set up, as shown in Figure 7. In the test scenario, the robotic fish is placed horizontally at a distance $d_{1}$ from the rear wall and is perpendicular to the rear wall. At this point, the edge pixels along the ξ direction, i.e., green points in Figure 7, are determined from the left and right images, and the real physical points corresponding to the two edge pixels are found on the rear wall. The horizontal distance between the real physical points, $d_{\Pi_{\zeta },2}$, is measured. The field of view range of the design plane of the field of view can then be calculated through the geometric relationship using the following equation:(8)$$\begin{array}{c} \Theta =360^{\circ }-2\arctan\left(\frac{d_{\Pi_{\zeta },2}}{2d_{1}}\right) \end{array}$$
After measurement, $d_{1}=30$ cm and $d_{\Pi_{\zeta },2}=75.5$ cm. Upon calculation, the deployed fish-like vision system has a field of view of 256.95°.

Through a similar method, the horizontal field of view is measured. The edge pixels corresponding to the horizontal plane direction, i.e., red points in Figure 7, and the corresponding real physical points, i.e., red points on the wall, are found. The distance between the real physical points is measured to be $d_{H,2}=30.20$ cm. After calculation, the horizontal field of view range is $\Theta_{H}=360^{\circ }-2\arctan\left(\frac{d_{H,2}}{2d_{1}}\right)$ = 306.5647°.

According to the experimental results, the field of view range $\Theta_{H}$ on the horizontal plane is greater than Θ on the field of view design plane. This is because the extent of binocular overlap is the greatest in the ξ direction on the field of view design plane, making the field of view angle in this direction the smallest. As the direction deviates from the ξ direction, the field-of-view range gradually increases until there is no overlap between the left and right fields of view. At this point, the field of view angle theoretically reaches its maximum fixed value of 2θ, at which point the images in the left and right eye fields of view are not continuous.

According to the field of view test results, the fish-like vision system can achieve a maximum visual perception capability of 306.56° in the underwater environment. This provides a novel ultra-wide field-of-view perception scheme for underwater robots, which is expected to enhance the perception ability and efficiency of robots in vision-based underwater operations.

In terms of the fish-like vision system, due to the loss caused by the water medium to the field of view of a single camera, the underwater perception range is smaller than that in the air, and the overlapping area of the binocular field of view becomes narrower. Therefore, for the design of fish-like vision system, the design of the overlapping angle of the field of view φ and the angle of the field of view Θ must take into account the refraction loss in the underwater environment plus allowing for a margin.

### 5.2. Visual Field Stitching Test

An underwater scene is set up, the angle of the robotic fish is adjusted, and images are captured with markers filling the overlapping area to obtain original images $I_{\mathrm{origin}}$. After calibrating the original images using OCamCalib v3.0, feature matching methods are performed on the calibrated images $I_{\mathrm{calib}}$. Figure 8a shows the results of left and right eye feature matching using the SIFT, SuperPoint, and proposed marker-assisted feature-enhanced matching methods.

The first two methods can detect a sufficient number of feature points, but the distortion of feature points in the left and right eyes is different, making it difficult for both methods to correctly pair the features. The proposed method, however, first obtains feature points under binocular vision through chessboard corner detection, then numbers the feature points based on the detected placement directions $o_{\mathrm{left}}$ and $o_{\mathrm{right}}$, and finally determines the feature point pairs by filtering the same numbers. Compared to classic feature detection algorithms, the proposed method successfully obtains correct feature point pairs in the case of large disparities and distortions in the left and right eyes with the assistance of markers. The marker-assisted feature-enhanced matching method is more suitable for feature-sparse underwater scenes and ensures the accuracy of pairing through the numbering strategy.

Based on the feature point matching pairs, the homography matrix is calculated, and the stitched image is obtained through camera parameter optimization and seam line optimization. The complete stitching process is shown in Figure 8b. According to the test, the visual field stitching algorithm is capable of restoring the complete chessboard grid image in the overlapping area. Through the visual field stitching algorithm, the fish-like vision system is able to output complete visual images without field of view loss.

### 5.3. Orientation Indicator Test

The orientation indicator test is conducted based on a 6×9 chessboard grid, with the images captured as shown in Figure 9a. By identifying the chessboard grid corners in the image, the pixel coordinates $p_{\left(i,j\right)}$ for each position $P_{\left(i,j\right)}$ are obtained. The horizontal distance *d* is set to 16 cm, and the chessboard grid unit length $d_{c}$ is 3.5 cm; thereby, we calculate the pitch and yaw angles corresponding to each position $P_{\left(i,j\right)}$ as shown in Table 1 and Table 2, respectively. By combining the corresponding pixel coordinate set $p_{\left(i,j\right)}$, partial contour lines are drawn in the stitched image, and the corresponding yaw and pitch angle scales are marked, with the results shown in Figure 9b.

According to the orientation indicator outcomes, the contour lines for the yaw scale and pitch scale are not oriented horizontally or vertically but are instead inclined at specific angles. This is due to the installation angle of the fish-like vision system being inclined relative to the vertical plane; if the vision system meets the condition ξ=0, the stitched image will have horizontal and vertical direction indicators. According to the pitch scale in Figure 9b, the upper view area (pitch angle less than zero) covers a larger range than the lower view area (pitch angle more than zero), indicating that the constructed vision system tends to observe the field of view above the robot.

For the wide field of view perception, yaw and pitch angle scales help to provide reference orientations for targets of interest within the image. When performing underwater tasks, the robotic fish can swim towards the target based on its reference orientation, achieving a function similar to visual serving.

### 5.4. Comprehensive Performance Test

Combining image stitching and orientation indicators, the comprehensive performance of the fish-like vision system was tested under different swimming patterns. Figure 10 shows image snapshots of the fish-like vision system with the robotic fish in horizontal swimming, diving, and rolling swimming states, respectively. As shown in Figure 10a, due to the installation angle of the fish-like camera, when the robotic fish swims horizontally, its stitched images capture the view looking upwards from underwater, while the images to the front, to the left, and to the right are mostly distributed around the periphery of the stitched image. During diving, the robotic fish’s posture is oriented towards the bottom of the pool, with the majority of the field of view being underwater images, and the output images reflect the field of view changes during diving, as shown in Figure 10b. In the rolling swimming pattern, the water–air interface line in the field of view continuously rotates, reflecting the rolling state of the robot, as shown in Figure 10c.

Through experimental testing, the comprehensive performance of the proposed fish-like vision system has been validated. By imitating the characteristics of biological fish’s eyes, it adapts to the shape of the robotic fish, providing a wide-ranging underwater visual perception capability. The advantages and practical implications are mainly reflected in three aspects. Firstly, its design without a waterproof shell increases its field of view in underwater scenarios. Secondly, through a binocular image stitching method with large distortion and disparity, the system can achieve an ultra-wide perception range (with a visual perception range exceeding 300°). Thirdly, compared to traditional vision systems that are installed horizontally or vertically, the designed vision system is more suitable for the shape of the robotic fish and can achieve a greater field of view with fewer hardware cameras. Given these characteristics, the fish-like vision system demonstrates universal applicability in underwater scenarios. This is particularly evident in tasks such as visual-based underwater search and maintenance, where the ultra-wide field of view significantly reduces the blind spots of underwater robots, increases the probability of detecting targets, and enhances the efficiency of robots when completing underwater tasks. The proposed vision system is expected to advance the visual perception capabilities of underwater robots and expand their application fields in these scenarios.

Our work provides a novel visual configuration scheme and a large-disparity image stitching algorithm. However, the fish-like vision system still has certain limitations. Firstly, the edges of the stitched image still have some distortion, which may affect the correct recognition of objects at the image edges. Secondly, when applying the proposed vision system to underwater bionic robots, the stability of the output images needs to be improved. Future research directions to address these limitations include, but are not limited to, the following aspects. Firstly, based on the visual data set captured by the proposed system, we can further enhance the training of high-level image algorithms to improve the accuracy of captured visual images for scenarios such as underwater target detection. Secondly, we can research real-time image stabilization methods for underwater bionic robots by cropping or stitching images. Thirdly, based on the large perception range of the proposed vision system, we can research a framework for visual-based underwater search methods to explore strategies for improving search efficiency. The fish-like vision system has great potential in underwater robots, especially in bionic robotic fish, and further research is expected to enhance the development of fish-like visual perception capabilities.

## 6. Conclusions

Inspired by the visual system of biological fish, we propose a fish-like vision system with a wide field of view that is suitable for deploying on underwater vehicles with fish-like streamlined shapes. Regarding the proposed fish-like vision system, this paper primarily encompasses four aspects. Firstly, the visual field design method for the fish-like vision system is presented, and we elucidate the relationship between streamlined shape features, field of view demands, and camera parameters. Secondly, based on a field design method, a fish-like vision system is constructed and deployed on a bionic robotic fish and uses a solution without waterproof compartments to avoid refraction loss of the field of view. Thirdly, in consideration of the characteristics of significant distortion and disparity in the system, a visual field stitching algorithm is designed to merge the images from binocular eyes, providing a foundation for applications such as target recognition algorithms. Finally, an orientation alignment method is devised to solve for the relative orientation between the robot and positions corresponding to visual image points, and yaw and pitch indicators are overlaid on the stitched image. Experimental results demonstrate that the proposed vision system possesses a wide-area perception capability of 306.56° and validate the effectiveness of the visual field stitching algorithm and the orientation alignment method. The experimental results indicate the practical applicability of the visual system in underwater robotics, and we offer an effective visual perception solution from a biomimetic perspective.

In the future, the fish-like vision system will be used for target recognition on underwater robots. Additionally, based on the complete wide-field perception image, research on electronic vision stabilization methods will be conducted in order to obtain stable perception video output through real-time cropping of the output image. Furthermore, vision-based searching strategies will be explored based on the advantage of wide-area visual perception to further enhance underwater search efficiency.

## Figures and Tables

**Figure 1 biomimetics-09-00171-f001:**
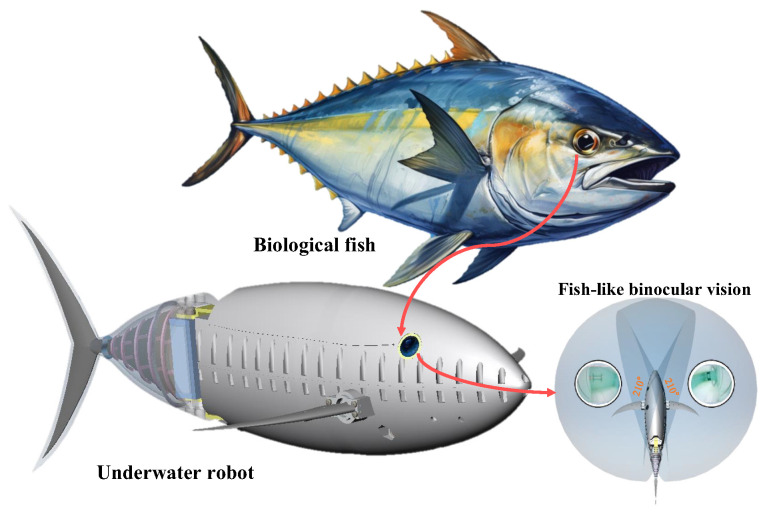
Schematic diagram of fish-like binocular vision system.

**Figure 2 biomimetics-09-00171-f002:**
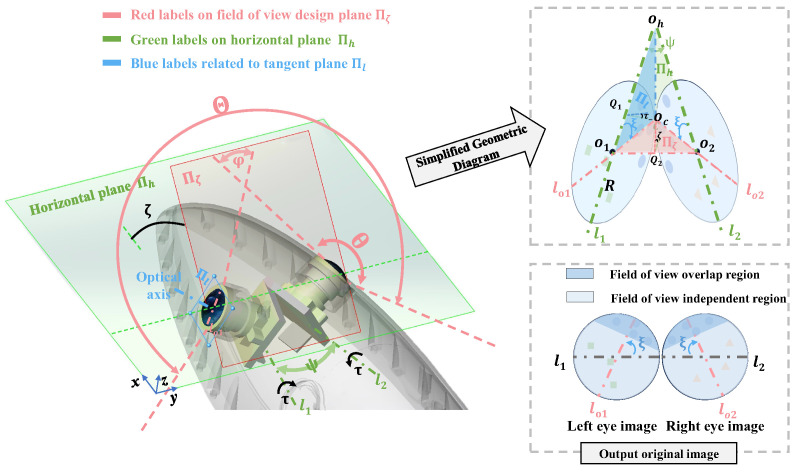
Field of view design: three key factors and camera mount geometry.

**Figure 3 biomimetics-09-00171-f003:**
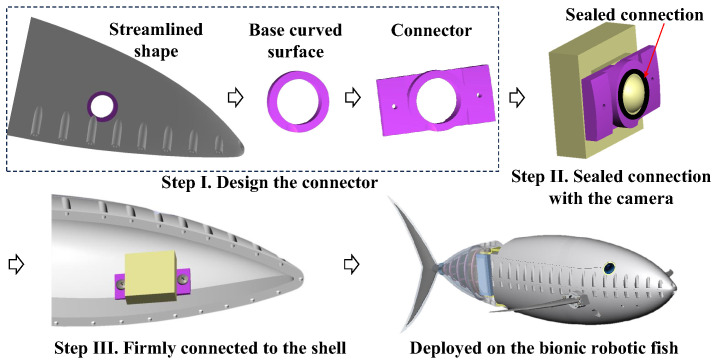
Deployment of fish-like vision systems in underwater environments.

**Figure 4 biomimetics-09-00171-f004:**
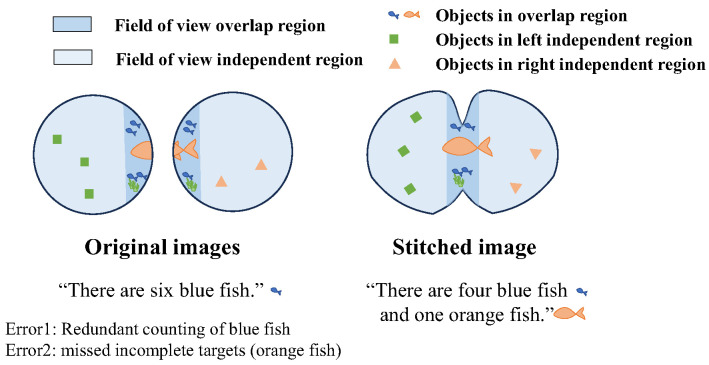
Stitched image helps to avoid ambiguity in target recognition in fish-like binocular vision.

**Figure 5 biomimetics-09-00171-f005:**
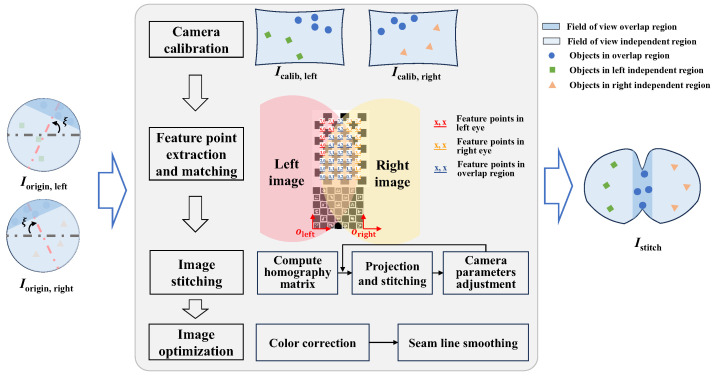
Process of binocular visual field stitching algorithm of fish-like vision system.

**Figure 6 biomimetics-09-00171-f006:**
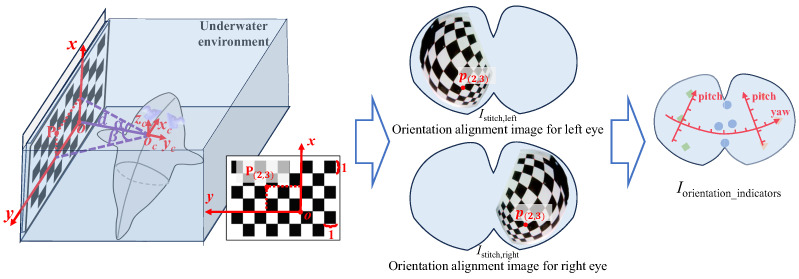
Orientation alignment of fish-like vision system.

**Figure 7 biomimetics-09-00171-f007:**
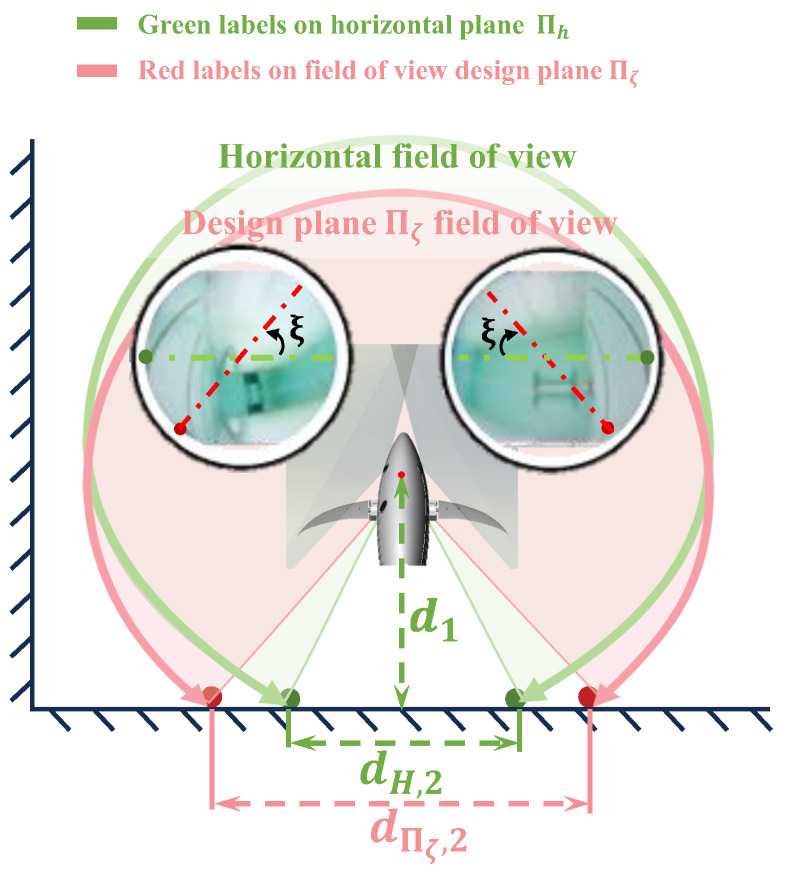
Experiment environment for field of view test.

**Figure 8 biomimetics-09-00171-f008:**
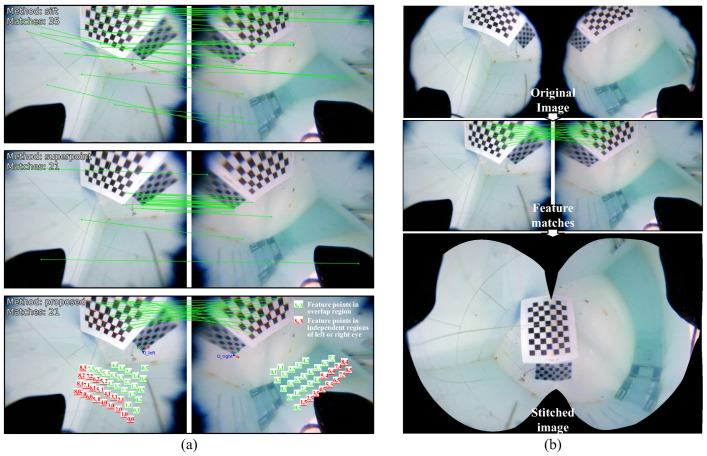
Stitching algorithm testing. (**a**) Comparative test of SIFT, SuperPoint, and proposed matching methods. (**b**) Based on original images, images featuring annotated feature points in the overlapping region are obtained through calibration and feature matching, ultimately resulting in the corresponding stitched image through the stitching algorithm.

**Figure 9 biomimetics-09-00171-f009:**
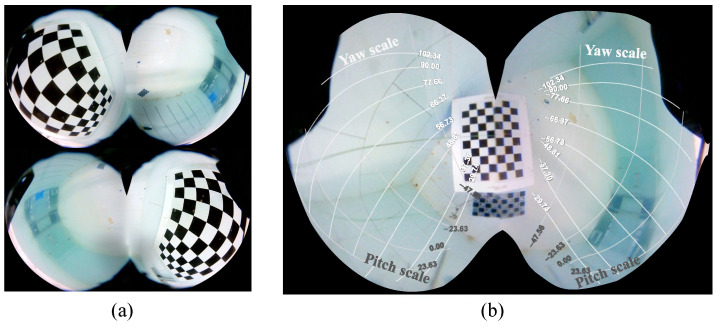
Results of orientation indicator testing. (**a**) Orientation alignment images captured by left and right eyes, respectively. (**b**) Stitched image with yaw and pitch scales.

**Figure 10 biomimetics-09-00171-f010:**
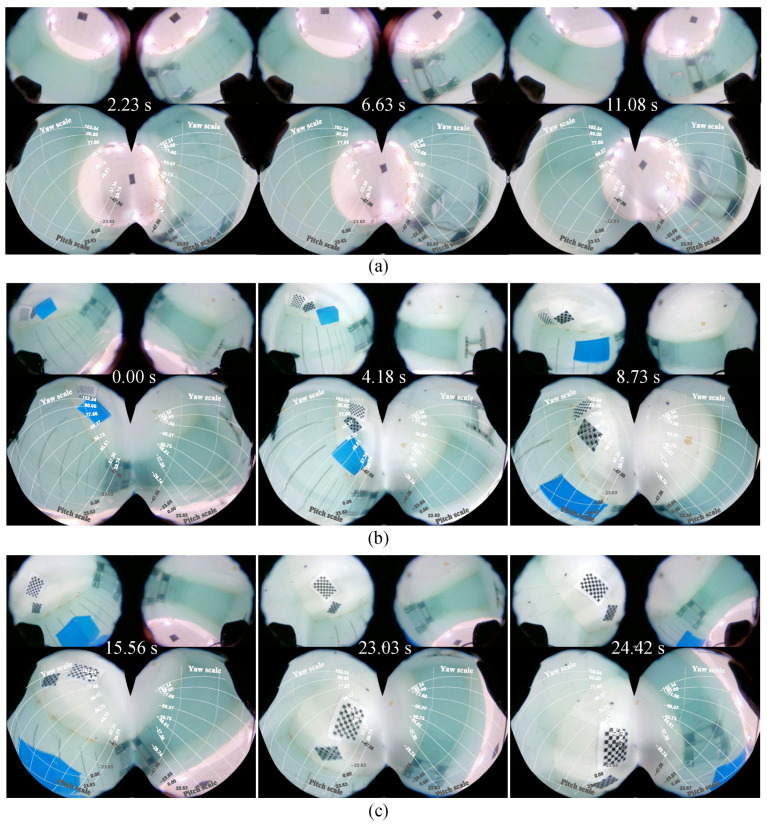
Comprehensive performance results. (**a**) Horizontal swimming. (**b**) Diving motion. (**c**) Rolling swimming.

**Table 1 biomimetics-09-00171-t001:** Pitch scale contour values.

i	−2	−1	0	1	2	3	4	5
Pitch scale (°)	23.63	12.34	0	−12.34	−23.63	−33.27	−41.19	−47.56

**Table 2 biomimetics-09-00171-t002:** Yaw scale contour values.

j	−2	−1	0	1	2	3	4	5	6	7	8
Yaw scale (Left) (°)	113.6	102.3	90	77.66	66.37	56.73	48.81	42.44	37.30	33.15	29.74
Yaw scale (Right) (°)	−113.6	−102.3	−90	−77.66	−66.37	−56.73	−48.81	−42.44	−37.30	−33.15	−29.74

## Data Availability

The data generated during the current study are available from the corresponding author on reasonable request.

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
