# Peer review of "A Fish-like Binocular Vision System for Underwater Perception of Robotic Fish"

_biomimetics, 2024, doi:10.3390/biomimetics9030171_

Round 1

Reviewer 1 Report

Comments and Suggestions for Authors
  1. The authors should motivate the readers to understand how difficult it is to design a fish-like binocular system for robotic fish. It is essential to establish it as a research problem. Specifically, how challenging is it to design and explore All three aspects of research present in the manuscript? For example, the field of view design and test description seems straightforward. However, I found the process of binocular visual field stitching algorithm more challenging and novel. Hence, it is tough for me to give the same weight to all three aspects from the research perspective. I would like to get more explicit opinions from authors to know their USP of this work, as the current version contains engineering design challenges and scientific experiments. 

  2. The statement mentioned in "Overlapping binocular fields of view" should be backed by scientific references. 

  3. The visibility of Figure 2 needs to be improved. 

  4. It would be more useful if the authors provided numerical examples of calculating matched feature point pairs in the feature point extraction and matching section. 

  5. How is the use of markers cost-effective? Please explain it and also provide scientific references. 

  6. I found no scientific results to describe the image stitching and optimization schemes. However, in the description of the image optimization section, the authors mentioned energy maps and Gaussian smoothing techniques. I assumed these statements were only discussed for theoretical backgrounds, and the authors performed these operations via camera parameter optimization. If that is the scenario, authors should add proper scientific references for these statements. However, stating how the authors optimized the camera parameters is essential. 

  7. The authors should highlight how this work will be used for target recognition.

Author Response

First of all, we greatly appreciate the valuable feedback. We have taken sincere care of the issues raised by you and thoroughly revised the manuscript. We highlighted the revised description in the manuscript. Please check the response to each comment in the attached response file.

Reviewer 2 Report

Comments and Suggestions for Authors

The article discusses the development of a fish-like binocular vision system for underwater perception in robotic fish. To improve the paper, the authors may consider the following suggestions:

  1. Clarify the field of view design method: Provide more details and explanations on the relationship between the streamlined shape features, field of view demands, and camera parameters.
  2. Explain the visual field stitching algorithm: Elaborate on the steps and techniques used for merging the images from binocular eyes and how it provides a foundation for target recognition algorithms.
  3. Provide more details on the orientation alignment method: Expand on the process of solving the relative orientation between the robot and positions corresponding to visual image points, particularly the overlaying of yaw and pitch indicators on the stitched image .
  4. Discuss the practical implications: Explain in more detail how the fish-inspired visual system can advance the visual perception capabilities of underwater robots and expand their application fields. Provide specific examples of potential applications and highlight the benefits.
  5. Include a discussion on limitations and future directions: Address the current challenges faced by underwater robot visual systems, such as image calibration complexity, limited visual perception range, and the need for vision stabilization methods. Additionally, suggest areas for further research and potential solutions to overcome these difficulties .
Comments on the Quality of English Language

The paper's english should be enhanced. 

Author Response

(The authors gave the same response as above.)

Round 2

Reviewer 1 Report

Comments and Suggestions for Authors

I want to thank the authors for such detailed responses. 

Reviewer 2 Report

Comments and Suggestions for Authors

No more comments 

Comments on the Quality of English Language

Minor grammatical errors exist.